# ELSA: LOCAL SPATIAL AUTOCORRELATION OF EMBEDDINGS

## ABSTRACT

Spatial autocorrelation is a popular concept in geography to measure the neighborhood dependencies of continuous random variables with geographic coordinates, such as temperature measures or house prices. However, in the real world, the data associated with a given location is often more complex than a simple scalar value, such as images or text. Here, we introduce **E**mbedding **L**ocal **S**patial **A**utocorrelation, a new statistic to measure spatial interdependencies of data embeddings at a given location. **ELSA** adapts and expands one of the most central concepts in geography for the age of AI and its central data structure: embeddings. We highlight the utility of **ELSA** as a measure of spatial homo- and heterogeneity. Focusing on image embeddings, we provide experiments on using **ELSA** to identify geographic clusters, outliers and for spatial analytics of model error terms. We also comment on further potential applications of **ELSA** and discuss the shortcomings of our approach.

## 1 INTRODUCTION

The first law of geography famously states that "All things are related but near things are more related" Tobler (1970). This core idea of geospatial data science is captured within the concept of spatial autocorrelation– and several metrics exist to quantify it. Most popular of these is the Moran's I, introduced originally by Moran (1950) as a global, dataset-level measure, and expanded into its local, observation-level version, refereed to as "Lisa" (local indicators of spatial autocorrelation), by Anselin (1995).

Moran's I statistics assess how a variable correlates with itself across space. The Global Moran's I provides an overall measure for the entire study area, indicating whether the pattern is clustered, dispersed, or random. The Local Moran's I, on the other hand, identifies clusters or outliers at specific locations. To give an illustrative example, let's consider the prices of houses within a city: a high Global Moran's I might indicate that high-priced homes are generally clustered together. Local Moran's I values can be used to pinpoint specific neighborhoods where this clustering is particularly strong or where there are anomalies, such as a high-priced home in a low-priced area. These metrics help geographers and practitioners like policymakers identify spatial patterns and e.g. target interventions.

Moran's I is limited to continuous random variables and a assumes a single, continuous scalar value (e.g. house price) for each location. As such, its use cases are limited to datasets that fulfill this requirement. In the real world, however, we often have more complex datatypes associated with a location, for example a social media post or a satellite image. These complex datatypes are often represented in vector embeddings, obtained via (pretrained) encoder models such as CLIP Radford et al. (2021). In this study, we develop a local spatial autocorrelation metric for data embeddings, allowing for the convenient analysis of spatial interdependencies of e.g. image and text data with geographic coordinates.

Our **E**mbedding **L**ocal **S**patial **A**utocorrelation (ELSA) metric is inspired by the local Moran's I, but leverages cosine similarities as a more suitable distance function between embedding spaces. Having a better intuition of spatial autocorrelation for images and text data can be useful in many different ways. For example, ELSA could help detect regions with similar or dissimilar land cover types from satellite imagery, helping to monitor environmental changes such as deforestation or urban sprawl. By identifying clusters of similar land cover, researchers can identify areas experiencing rapid change or anomalies, such as unexpected vegetation loss in a typically forested area. ELSA might further be used to analyze predictive models and their performance in different geographic areas, identifying areas of relative spatial heterogeneity as more difficult to classify than areas of relative spatial homogeneity.

This study details the design of the ELSA metric and highlights its computation and use on real-world image datasets. Our contributions can be summarized as follows:

- We propose **ELSA**, a first of its kind measure of local spatial autocorrelation for data embeddings.
- We introduce a permutation testing regime allowing for seamless testing of statistical significance of the **ELSA** metric.
- Across several different image datasets, We run experiments highlighting the utility of **ELSA** for identifying geographic clusters of homogeneous data, detecting outliers, and for identifying difficult-to-predict data samples.

## 2 RELATED WORK

### 2.1 SPATIAL AUTOCORRELATION

Moran's I, introduced by Patrick A. P. Moran in 1950 Moran (1950), is one of the earliest and most widely used measures of spatial autocorrelation. It quantifies the degree to which a variable is similarly distributed across space, capturing the correlation between a variable at one location and the same variable at neighboring locations. Negative values of the statistic imply spatial dispersion, while positive values indicate clustering, and values around zero indicate spatial randomness. Moran's I laid the foundation for spatial statistics by formalizing the concept of spatial dependence in quantitative terms, and it has since become an essential concept in academic disciplines such as econometrics Jin & Lee (2015), geology Tepanosyan et al. (2019), and ecology Diniz-Filho et al. (2003).

While Moran's I is the most popular, there exist several other, related metrics for spatial autocorrelation: Geary's C Geary (1954), inversely related to the Moran's I, computes spatial autocorrelation using sum of squared distances, as opposed to Moran's I use of standardized spatial covariances. Other notable methods include the Getis-Ord statistics Getis & Ord (1992) for hotspot analysis. Just like the Moran's I, the Getis-Ord statistic also has been expanded into a localized version Ord & Getis (1995). Recent years have seen the emergence of expansions of Moran's I and other spatial autocorrelation metrics to spatio-temporal data Shen et al. (2016); Klemmer et al. (2022) and multivariate data Anselin (2019); Lin (2023); Yamada (2024). However, existing multivariate spatial autocorrelation statistics all rely on notions of Euclidean distance and are thus less applicable to embedding spaces.

### 2.2 MACHINE LEARNING FOR & WITH GEOGRAPHIC DATA

As with other domains, machine learning is becoming increasingly popular for geospatial data, owing especially to its high scalability. Impactful applications of modern machine learning and deep learning techniques in the area include for example land-cover mapping Rußwurm et al. (2020), monitoring of global fishing activities Paolo et al. (2024) or carbon stock estimation Reiersen et al. (2022). Geospatial data, however, also comes with distinct characteristics such as spatio-temporal dynamics, resolution sensitivity or high number

of spectral bands. This has prompted calls for dedicated, methodological research into machine learning techniques tailored to geospatial data Reichstein et al. (2019); Rolf et al. (2024).

Recent years have seen emerging work in geospatial machine learning, a discipline that fuses concepts from geography and machine learning into purpose-built tools. In this line of work fall studies proposing geospatial expansions of popular machine learning algorithms, such as geospatial random forest Geerts et al. (2024), studies adapting popular geographic concepts for modern machine learning Liu et al. (2022), and studies proposing new architectures and methods incorporating geospatial structures Zammit-Mangion et al. (2022). The Moran's I metric has itself been used successfully within machine learning algorithms, for example as auxiliary task in predictive modeling Klemmer & Neill (2021), or as embedding loss function in generative models Klemmer et al. (2022). Nonetheless, current research leaves a concrete gap: The expansion of one of the central concepts of geographic data, spatial autocorrelation, to the data format of the AI age–embeddings. We aim to address this gap in the current study.

## 3 METHOD

### 3.1 LOCAL SPATIAL AUTOCORRELATION

The local Moran's I measure of spatial autocorrelation is defined as:

$$I_i = \frac{z_i}{s_2} \sum_{j=1}^{n} w_{ij} z_j$$

where $I_i$ is the local Moran's I for location $i$, $z_i = y_i - \bar{y}$ is the deviation of the value at location $i$ from the mean $\bar{y}$, $s_2 = \frac{1}{n} \sum_{i=1}^{n} (y_i - \bar{y})^2$ is the variance, $w_{ij}$ is the spatial weight between locations $i$ and $j$, and $n$ is the number of locations. The local Moran's I metric depends on a notion of spatial adjacency, captured by the weight matrix $W = [w_{ij}]$. This matrix can be obtained via nearest neighbor search and may be binary or weighted by the distance between two locations.

### 3.2 EMBEDDING LOCAL SPATIAL AUTOCORRELATION

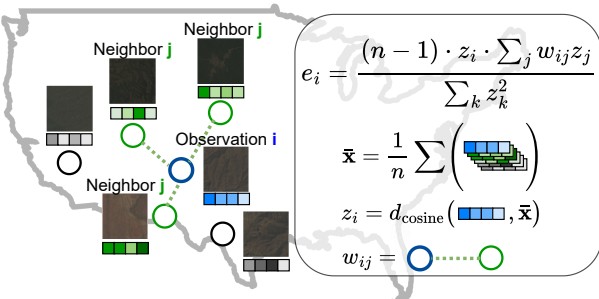

Figure 1: **Computing the ELSA metric of an image embedding:** Image embeddings **x** are extracted from the images and mapped onto a graph with adjacency matrix $W$, defined by spatial neighborhood. The ELSA scores $e$ for each embedding are then constructed as

To generalize spatial autocorrelation to high-dimensional embeddings, we develop the Embedding Local Spatial Autocorrelation (ELSA) measure. We assume as input embedding vectors $\mathbf{x}_i \in \mathbb{R}^d$ of dimension $d$ and a spatial weight matrix $W = [w_{ij}]$.

**Definition 1** (ELSA Measure). The ELSA measure $e_i$ for an embedding $\mathbf{x}_i$ is defined as:

$$e_i = \frac{(n-1) \cdot z_i \cdot \sum_j w_{ij} z_j}{\sum_k z_k^2}$$

where $z_i$ represents the standardized cosine similarity between embedding $\mathbf{x}_i$ and the global mean embedding $\bar{\mathbf{x}}$. $i$ gives the index of the current data point, $j$ the index of neighbors of $i$, and $k$ indexes the all observations.

The standardization of cosine similarity is given by:

$$z_i = \frac{\mathbf{x}_i \cdot \bar{\mathbf{x}} - \mu_{\cos}}{\sigma_{\cos}}$$

where $\mu_{\cos} = \frac{1}{n} \sum_{k=1}^{n} (\mathbf{x}_k \cdot \bar{\mathbf{x}})$ is the mean cosine similarity, $\sigma_{\cos} = \sqrt{\frac{1}{n} \sum_{k=1}^{n} (\mathbf{x}_k \cdot \bar{\mathbf{x}} - \mu_{\cos})^2}$ is the standard deviation of cosine similarities, and $\bar{\mathbf{x}} = \frac{1}{n} \sum_{i=1}^{n} \mathbf{x}_i$ is the mean embedding vector. For cases where $\sigma_{\cos} = 0$ (i.e., all embeddings are identical or perfectly aligned with the mean), we define $z_i = 0$ for all $i$.

**Lemma 1** (ELSA Range). Local ELSA values have no fixed bounds and can theoretically range from $-\infty$ to $+\infty$, even with row-standardized weights. In practice, most values typically fall within a moderate range, with:

- Strongly positive values, indicating that an embedding is surrounded by similar embeddings within a homogenous cluster.

- Values near zero, indicating no quantifiable spatial pattern.

- Strongly negative values, indicating that an embedding is surrounded by dissimilar embeddings who are similar among each other.

Intuitively, $e_i$ measures local spatial autocorrelation by considering the similarity of observation $i$ to its neighbors $j$, weighted by the spatial adjacency matrix $W$.

### 3.3 PERMUTATION TESTING

To assess the statistical significance of observed ELSA values, we employ permutation testing to compare the observed values against a distribution generated under the null hypothesis of complete spatial randomness.

**Definition 2** (Null and Alternative Hypotheses).

- **Null Hypothesis** ($H_0$): No spatial autocorrelation exists in the embedding space; the observed ELSA values $e_i$ are not significantly different from what would be expected under random spatial arrangement.
- **Alternative Hypothesis** ($H_1$): Significant spatial autocorrelation exists in the embedding space; the observed ELSA values $e_i$ differ significantly from random spatial arrangement.

**Lemma 2** (ELSA Expectation under $H_0$). Under the null hypothesis of spatial randomness, the expected value of $e_i$ is:

$$E(e_i | H_0) = 0$$

*Sketch.* Under random spatial arrangement, there is no systematic relationship between an embedding and its neighbors. The standardized $z$-scores and their spatial lags are uncorrelated, leading to an expected value of zero. □

The variance of $e_i$ under the null hypothesis can be estimated from the distribution of permuted values:

$$\text{Var}(e_i|H_0) = \frac{1}{n_{\text{iter}}} \sum_{p=1}^{n_{\text{iter}}} (e_i^{(p)} - E(e_i^{(p)}))^2$$

where $e_i^{(p)}$ are the permuted ELSA values obtained from the $p$-th permutation sample.

### 3.3.1 PERMUTATION PROCEDURE

The permutation test procedure is as follows:

1. Calculate the true ELSA values $e_i$ using the original spatial weight matrix $W$.
2. For each permutation $p \in \{1, 2, \ldots, n_{\text{iter}}\}$:
   (a) Randomly shuffle the locations of embeddings while keeping the spatial weight matrix $W$ fixed. This effectively breaks any spatial dependence between embeddings.
   (b) Compute the permuted ELSA values $e_i^{(p)}$ using the original $W$ and permuted embedding locations.
3. For each location $i$, calculate the empirical p-value:
$$p_i = \frac{1}{n_{\text{iter}}} \sum_{p=1}^{n_{\text{iter}}} \mathbb{I}(|e_i^{(p)}| \geq |e_i|)$$
where $\mathbb{I}$ is the indicator function that equals 1 if the condition is true and 0 otherwise.

A low p-value indicates that the observed ELSA value $e_i$ is unlikely to have occurred by chance, suggesting significant local spatial autocorrelation of the embeddings at location $i$.

**Lemma 3** (Asymptotic Distribution). For sufficiently large samples, under the null hypothesis, the standardized ELSA values approach a standard normal distribution:

$$\frac{e_i - E(e_i|H_0)}{\sqrt{\text{Var}(e_i|H_0)}} \sim N(0,1)$$

This asymptotic property allows for parametric inference when the number of permutations is computationally prohibitive.

## 4 EXPERIMENTS

To build a deeper intuition for what the ELSA metric captures, we show its application to several real- world datasets of images with associated geo-locations. We first provide a visual analysis, plotting the ELSA metric on the world map, highlighting areas of spatial homogeneity and heterogeneity. We also provide examples for images (and their neighbors) with low and high ELSA values, showcasing the metrics ability to identify samples similar or dissimilar from their surroundings. We then address the question whether ELSA embeddings can be indicative of the performance of a predictive model at a given location, i.e. whether embeddings at some locations are harder to classify.

### 4.1 DATASETS

We explore the ELSA metric and its utility on several real-world datasets. These can be split into two categories: labeled and unlabeled datasets.

**Labeled datasets:** As labeled datasets we chose four different datasets of locations with associated satellite image embeddings paired with environmental labels. These datasets from the Mosaiks project Rolf et al. (2021) are accessed via the TorchSpatial benchmark Cao et al. (2024) and include *Population*, *Elevation*, *Forest Cover* and *Nightlights* data. The predictive modeling task associated with these datasets is to predict a continuous outcome variable (population, elevation, forest cover or nightlights) $y$ from image vector embeddings $\mathbf{x}$ and location coordinates $\mathbf{c} = [\text{longitude}, \text{latitude}]$. Image embeddings are obtained using random convolutional features from global Planet satellite imagery.

**Unlabeled datasets:** As unlabeled datasets, we chose two dataset of natural images: IM2GPS3k is a dataset of internet images with geographic references curated by (Vo et al., 2017) and a subset of the larger IM2GPS dataset Hays & Efros (2008; 2014). YFCC4K is a subset of the YFCC100M dataset Thomee et al. (2016) of social media imagery extracted from Flickr, again curated by (Vo et al., 2017). Both of these datasets are commonly used as unseen test sets in geolocalization research.

## 4.2 VISUAL ANALYSIS

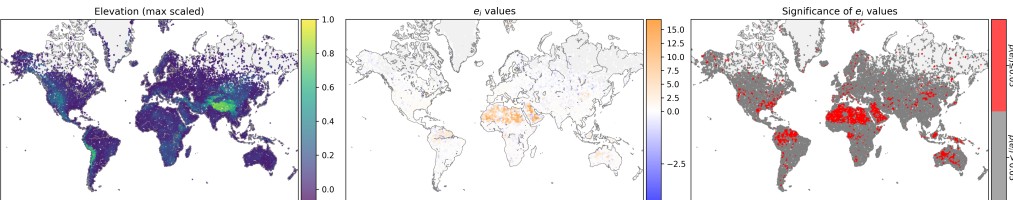

Figure 2: **Overview of the Elevation dataset:** The left panel shows the elevation values at all locations, the center panel shows the values of the ELSA metric computed for the satellite image embeddings at all locations and the right panel shows the statistical significance of the ELSA metrics from the center panel.

As a first step in our experiments, we conduct a visual analysis of the ELSA statistic on our datasets to help build an intuition of how the metric manifests.

Figure 2 gives an overview of the ELSA statistic for the Mosaiks Elevation dataset. From left to right, the figure shows world maps of elevation values, ELSA values of satellite image embeddings and statistical significance of ELSA values. We can observe high autocorrelation, i.e. spatial homogeneity, in larger areas with shared visual features, such as the Sahara desert or the Amazon rain forest. Negative autocorrelation indicates that the visual features of a location are starkly different to a homogeneous surrounding area. This could for example be a city in the desert or the forest, such as Phoenix in the US or Manaus in Brazil. ELSA values around zero indicate that there is no clear spatial dependency present in the image embeddings of a given location.

Figure 3 provides example images from the IM2GPS3K dataset associated with large negative and positive ELSA values. In 3a we can see a negatively autocorrelated image which contrasts starkly from its homogenous neighbors, while in 3b we can see an image very similar to all of its neighbouring images. For images with ELSA scores around zero, there would not be an observable clear relationship, nor positive not negative.

## 4.3 ANALYZING IMAGE REGRESSION RESIDUALS

A common use case for the local Moran's I is to help understand the error term of predictive models. Following this idea, we seek to compare the ELSA values of image embeddings to the error term of predictive models that use these image embeddings as input. To start, we focus on image regression task on the four

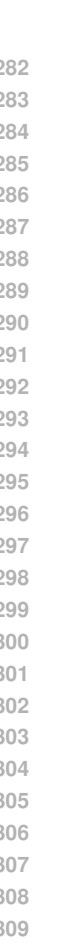
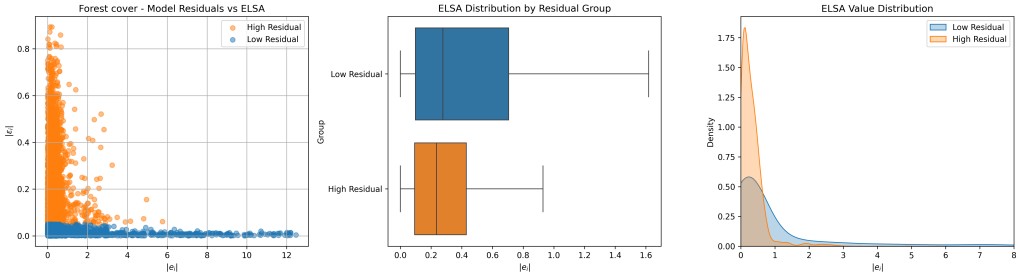

(a) Example for a geo-referenced image with strong negative spatial autocorrelation and its neighbors.

(b) Example for a geo-referenced image with strong positive spatial autocorrelation and its neighbors.

Figure 3: **Examples from the IM2GPS3K dataset:** The left figure shows an example for an image with a large negative ELSA score, indicating negative autocorrelation. The right figure shows an example for an image with large positive ELSA score, indicating positive autocorrelation.

Figure 4: **Image regression residual analysis on Mosaiks Forest Cover:** The left panel shows a scatterplot of the absolute model errors $|\epsilon|$ versus absolute ELSA values $|e|$, the center panel show boxplots for the distribution of absolute ELSA values, split by high versus low absolute error, the right panel shows a kernel density estimation of the distribution of absolute ELSA values for high and low error residuals.

Mosaiks datasets: elevation, population, nightlights and forest cover prediction. For each of these tasks, we train simple fully-connected neural networks that aim to predict outcomes $y$ from image embeddings $\mathbf{x}$ and location coordiantes $\mathbf{c}$: $y \sim f_{\text{Pred.Model}}(\mathbf{x}, \mathbf{c}) + \epsilon$. We then take the test set error term $\epsilon$ from the trained predictive models and compare it to the ELSA scores $e$ of the image embeddings.

Figure 4 shows a detailed analysis for the forest cover image regression task. In this figure, we split our test set into observations with high and low absolute error term and analyze how the distribution of ELSA values differs between these groups. We can observe a clear trend: images whose forest cover is harder to predict–i.e. that have a larger absolute error–are associated with lower absolute ELSA values–i.e. are less autocorrelated. This implies that images with higher autocorrelation are easier to predict. This finding is somewhat intuitive, as autocorrelated observations can be easier interpolated using neighbouring observations, if there is a stronger neighbourhood relationship.

Table 1: **Results from Welch's t-test for ELSA values split by image regression errors:** Absolute regression residuals are split into low and high residual groups ($|\epsilon| <= 0.05$ and $|\epsilon| > 0.05$ respectively). Welch's t-test is used to compare the means of absolute ELSA values $|e|$ in both groups for statistically significant difference.

| **Dataset** | **Low $\|\epsilon\|$ Group** | | | **High $\|\epsilon\|$ Group** | | | **t-statistic** | **p-value** |
|---|---|---|---|---|---|---|---|---|
| | Count | Mean | Median | Count | Mean | Median | | |
| Elevation | 2585 | 0.856 | 0.273 | 2396 | 0.495 | 0.242 | 8.501 | 0.000*** |
| Population | 1240 | 0.878 | 0.267 | 3016 | 0.645 | 0.254 | 3.712 | 0.000*** |
| Forest Cover | 1974 | 1.157 | 0.277 | 3007 | 0.331 | 0.234 | 15.987 | 0.000*** |
| Nightlights | 4394 | 0.730 | 0.240 | 528 | 0.336 | 0.289 | 13.545 | 0.000*** |

Statistical significance: $^{*}p < 0.05$, $^{**}p < 0.01$, $^{*}p < 0.001$

To expand on this analysis, we run a Welch t-test Welch (1947) to compare the ELSA values of low and high error groups. The Welch t-test is a statistical test used to determine whether the means of two groups are significantly different. It is commonly used for comparing two samples without equal sample size and without assumed equal variance. Our results, presented in Table 1, show that across all our image regression datasets ELSA values are significantly different between low and high error groups. This confirms our hypothesis that predictions are easier when the input images have stronger (absolute) autocorrelation.

## 4.4 ANALYZING GEOLOCALIZATION ERRORS

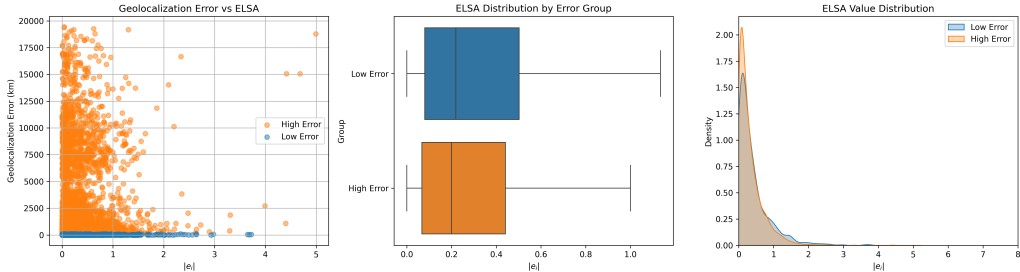

Figure 5: **Geolocalization error analysis on YFCC4K:** The left panel shows a scatterplot of the geolocalization errors $|\epsilon|$ versus absolute ELSA values $|e|$, the center panel show boxplots for the distributions of absolute ELSA values, split by high versus low geolocalization errors, the right panel shows a kernel density estimation of the distribution of absolute ELSA values for high and low geolocalization error groups.

We now focus our analysis on a different predictive modeling problem involving geographic coordinates: image geolocalization. Here, the task is to predict for a given image embedding $\mathbf{x}$ its geographic coordinates $\mathbf{c}$: $\mathbf{c} \sim f_{Geoloc.Model}(\mathbf{x})$. In practice, rather than directly predicting longitude and latitude coordinates from an image input, this problem is often discretized, either by gridding the planet and having the localization model predict the correct grid cell Vo et al. (2017), or through CLIP-style location-image matching Vivanco Cepeda et al. (2023); Klemmer et al. (2025).

Here we use the later approach and aim to predict the locations of images in the IM2GPS3K and YFCC4K datasets using a pretrained GeoCLIP Vivanco Cepeda et al. (2023) model. Note that this model has never seen the images in these datasets. Figure 5 adapts the analysis of Figure 4 to geolocalization errors of

Table 2: **Results from Welch's t-test for ELSA values split by geolocalization errors:** Geolocalization errors are split into low and high error groups ($|\epsilon| \leq 100km$ and $|\epsilon| > 100km$ respectively). Welch's t-test is used to compare the means of absolute ELSA values $|e|$ in both groups for statistically significant difference.

| Dataset | Low $|\epsilon|$ Group | | | High $|\epsilon|$ Group | | | t-statistic | p-value |
|---------|-------|------|--------|-------|------|--------|-------------|---------|
|         | Count | Mean | Median | Count | Mean | Median |             |         |
| IM2GPS3K | 1229 | 0.356 | 0.190 | 1768 | 0.462 | 0.260 | $-5.245$ | 0.000*** |
| YFCC4K   | 1185 | 0.390 | 0.220 | 3351 | 0.327 | 0.199 | 4.064 | 0.000*** |

Statistical significance: $^*p < 0.05$, $^{**}p < 0.01$, $^{***}p < 0.001$

GeoCLIP on the YFCC4K dataset. We can again observe that observations with lower geolocalization error are characterized by higher absolute ELSA values, implying that spatially autocorrelated images are easier to geolocalize.

These results hold under the Welch t-test scheme, as Table 2 shows. However, we do not find these results to hold for our other dataset, IM2GPS3K. In fact, for this dataset we observe the inverse relationship, with observations with lower geolocalization error being associated with lower autocorrelation scores. Overall, this shows that the relationship between image embedding ELSA values and predictive performance is more robust for image regression tasks than for image geolocalization tasks.

## 5 CONCLUSION

### 5.1 DISCUSSION OF LIMITATIONS AND RESEARCH CHALLENGES

Before concluding this study, we discuss the limitations of our work and comment on new research challenges arising from our findings. One of the key shortcomings of the Moran's I measure also applies to our ELSA metric: its dependence on a predefined spatial adjacency matrix $W$. In cases where no predefined adjacency is available and e.g. geographic coordinates are used for creating a kNN graph, this might require some a-prior understanding of the spatial dependencies of the data at hand. While our experiments focus on image embeddings, the ELSA metric can be applied to arbitrary data embeddings and may e.g. be used to analyze autocorrelation in textual data. We hope to explore this avenue in future work.

Our study also enables work expanding intuitions of embedding autocorrelation and their use in machine learning. One impactful direction for future work is the expansion of ELSA for spatio-temporal data (e.g. images with both geographic coordinates and time stamps), inspired by spatio-temporal expansions of the Moran's I Shen et al. (2016). Further research could explore uses of ELSA for improving predictive and generative modeling of geospatial data, inspired by similar uses of the Moran's I metric Klemmer & Neill (2021).

### 5.2 CONCLUSION

In this study we propose **ELSA**, a new measure of spatial autocorrelation for embeddings. This adapts a central measure of geographic dependencies to flexible embedding data structures, allowing for the analysis of complex data such as images or text with geographic references. **ELSA** enables the identification of geographic clusters and outliers in complex data types, as we highlight in our experiments on several image datasets. Our study enables several promising research directions, from the sampling of geographically diverse datasets to the integration into supervised and self-supervised learning settings.

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

## STATEMENT ON LLM USAGE

LLMs have been used to polish and simplify writing, to check for spelling and grammatical errors, and to refactor the code provided in the supplementary materials. They have not otherwise been used in this work.

