# OpenReview forum: "ELSA: Local spatial autocorrelation of embeddings"
_ICLR.cc/2026/Conference — ICLR 2026 Conference Withdrawn Submission_

### Official Review · Reviewer_AadM · 2025-10-25

**Soundness:** 2
**Presentation:** 2
**Contribution:** 1
**Rating:** 2
**Confidence:** 4

**Summary:**

This paper introduces ELSA, the first local spatial autocorrelation measure for high-dimensional embeddings, extending geography's classical Local Moran's I from scalars to vectors such as images or texts. By quantifying how similar an embedding is to its neighbors via cosine similarity and providing permutation-based significance testing, ELSA visually uncovers homogeneous regions like deserts and forests and pinpoints anomalies such as cities in the Sahara. Across satellite and Internet photo datasets, predictive errors are markedly lower in high-ELA, strongly autocorrelated areas, offering geospatial AI a new lens for spatial analysis and model diagnostics.

**Strengths:**

1. This paper introduces the ELSA metric, which is inspired by the local Moran’s I but leverages cosine similarities as a more suitable distance function between embedding spaces.
2. They also introduce a permutation testing regime allowing for seamless testing of the statistical significance of the ELSA metric.
3. They conduct experiment verification on multiple image datasets, demonstrating the practicality of ELSA.

**Weaknesses:**

1. The novelty of this paper is far from the standard of ICLR. Simply proposing a metric and verifying it is not sufficient to become a top-conference paper. It is suggested that the author make the work more complete before submission (e,g,, more contribution is needed; the number of experiments is quite limited ......).

2.  Section 4.2 Visual analysis only provides a demonstration of the dataset used in experiments, instead of providing visualization analysis of experiment results.

2. Regarding the experiments, I think it lacks corresponding baseline models. As stated before, the number of experiments is limited, and no extra experiment content is provided in the appendix as well.

  The author stated that ELSA could help detect regions with similar or dissimilar land cover types from satellite imagery. Then they should provide some intuitive presentation of the effectiveness of ELSA, instead of merely reporting the error analysis.

4. The paper lacks a detailed introduction for the model framework, such as image encoder.
In line 319, how is the image embedding x generated?

**Questions:**

see weakness.

---

### Official Review · Reviewer_E3vJ · 2025-10-28

**Soundness:** 1
**Presentation:** 1
**Contribution:** 1
**Rating:** 0
**Confidence:** 3

**Summary:**

The paper proposes ELSA, a local spatial autocorrelation measure for geospatial embeddings. The authors repurpose Moran's I metric to enable computing spatial autocorrelation for high dimensional geospatial embeddings. ELSA metric could be used to understand the spatial variance of geospatial embeddings which can in turn reflect the predictive power of embedding models.

**Strengths:**

1. The paper proposes Moran's I metric for high-dimensional geospatial embeddings. This metric could be used to understand clusters and task performance.

**Weaknesses:**

1. Limited technical novelty and unclear contribution in terms of future applications of ELSA in geospatial applications.
2. The writing needs to be improved a lot. Many sentences seem like GPT-generated.
3. Can this framework be only used for regression tasks? I would like to see some analysis on classification tasks such biome/ecoregion classification or land-cover classification.
4. Can ELSA be made more task-aligned? Currently it seems for a given model, the ELSA values will be the same irrespective of the downstream task. I think ELSA needs to be made more task-specific since some tasks are spatially smooth e.g. temperature prediction while some tasks are not e.g. geolocalization.
5. What would happen if geolocation embedding is concatenated along with the image embeddings? How would the ELSA metric change?
6. Can you discuss if the ELSA metric is influenced by the spatial distribution of training data? For example, most of the datasets used in experiments are biased towards North America and Europe. Is ELSA robust to domain shifts?
7. Can you compare other location embedding models and satellite embedding models? The paper is limited to Mosaiks and GeoCLIP embeddings. Also, how would ELSA metric change when using high-resolution satellite imagery vs low-resolution satellite imagery?
8. After computing ELSA metric, how can one improve the embedding models for a particular downstream task?

**Suggestions**:

I think the paper presents an interesting analysis but is very limited in terms of the depth of experiments. The authors need to rethink how ELSA metric could be made more task-specific as currently ELSA is fixed given a particular embedding model. The authors need to show concrete applications of the metric rather than just providing an analysis that compares ELSA with downstream performance.

**Questions:**

Please see weaknesses.

---

### Official Review · Reviewer_c1in · 2025-10-29

**Soundness:** 3
**Presentation:** 3
**Contribution:** 2
**Rating:** 4
**Confidence:** 4

**Summary:**

This paper proposes ELSA (Embedding Local Spatial Autocorrelation), a metric for quantifying spatial dependence among high-dimensional embedding vectors tied to geographic locations. ELSA replaces scalar values with embeddings and uses cosine similarity to measure how each point relates to the global mean embedding. Weighted by a spatial adjacency matrix, it produces local scores that capture spatial homogeneity or heterogeneity in the embedding space. Experiments show that ELSA can detect geographic clusters, identify outliers, and reveal areas where predictive models perform poorly, illustrating its potential as a diagnostic tool for spatial embedding analysis.

**Strengths:**

1. The authors extend local spatial autocorrelation to high-dimensional embedding spaces, which is an interesting and important problem in AI-based geospatial analysis.

2. The paper is clearly written and easy to follow.

3. The experiments address practical scenarios such as detecting homogeneous geographic clusters and identifying spatial outliers, demonstrating the potential value of the proposed framework in real-world applications.

**Weaknesses:**

1. The paper does not analyze how hyperparameter choices such as the distance metric and spatial weighting matrix influence the computed ELSA scores. Because these factors directly determine the magnitude and stability of spatial correlation, the absence of such analysis raises concerns about the interpretability and robustness of the results.

2. The evaluation is limited to embedding representations of satellite and geographic images. Although the paper claims that the proposed metric has broad applicability, there is no evidence demonstrating its generalizability to other domains, which limits the strength of the contribution.

3. The paper lacks a comprehensive discussion of related work that could potentially address similar problems, and the experimental section does not include comparisons to existing baselines, which weakens the empirical justification of the proposed method.

4. The novelty of the proposed method is limited. It seems to be a straightforward extension of Moran’s I to the embedding space.

**Questions:**

Please see weaknesses

---

### Official Review · Reviewer_uwZP · 2025-10-30

**Soundness:** 2
**Presentation:** 3
**Contribution:** 2
**Rating:** 4
**Confidence:** 4

**Summary:**

## Problem:
- Spatial data analysis is becoming increasingly high-dimensional as complex data modalities such as image and text are used to characterize spatial phenomena.
- Historically spatial autocorrelation has been a useful statistic to characterize spatial data, however such statistics are limited to scalar-valued random variables
- A new set of spatial autocorrelation are needed to age of high dimensional data AI models

## Solution:

- The authors propose the Embedding Local Spatial Autocorrelation (ELSA) statistic, that extends the idea of local spatial autocorrelation to learned embeddings produced by deep learning models
- Additionally, they introduce a non-parametric permutation test to facilitate geospatial hypothesis testing across distributions of embeddings
- They demonstrate that elsa can identify geographic clusters across a variety of of geospatial image datasets.

**Strengths:**

- A well written paper, an important problem and interesting idea.
- The presentation of the permutation is useful and pragmatic solution for hypothesis testing
- Interesting experiments showing that ELSA is predictive of geospatial regression model residuals

**Weaknesses:**

- The range of ELSA is unbounded. This can limit interpretability and can make decision-making from this statistic more difficult. See for example, the skewness in Figure 2b. On the positive side, ELSA values range up to 15, but on the negative side only down to -2 or -3.
- Lacking some theoretical rigor. None of the lemmas are presented with Proof. Lemma 3 for example, could likely benefit from more theoretical backing.
- No study of the  sensitivity of ELSA to the choice of encoder. For example, GeoCLIP vs DinoV3?

**Questions:**

- Figure 1 (line 136) - is this caption missing? It appears to end in an incomplete sentence?
- If all embeddings are identical and $z_i$ is set to 0, wouldn’t $z_k$ also be 0? Would this result in an undefined result due to the division by 0?
- Why is (n-1) introduced into the numerator of Definition 1?
- I am a little confused by the notation in Definition 1. Does $\boldsymbol{x} \cdot \boldsymbol{\bar{x}}$ indicate the dot product? Or the cosine similarity? I realize they are related, but not exactly equivalent. These could be made more clear in the paper
- In lemma 3, the authors argue that the ELSA statistic approaches a standard normal distribution. Is the permutation test necessary in light of this result?
- Figure 4/5: The ELSA values between high and low residual groups have significant overlap. It’s not clear to me that ELSA values could be used to distinguish between them. How is the p-value of the welch test so low, but the simple density plots are very similar? The results in Table 1 and Table 2 don’t quite make sense to me given the plot.

---

### Note · Authors · 2025-12-02

**Comment:**

We have decided to withdraw the paper given that the reviewers do not think that the paper is in a state warranting acceptance.
However, we want to thank all reviewers for their feedback and excellent comments and their general encouragement and positive comments on the topic of the paper and the proposed metric. The reviewer feedback will help us improve the paper for a future submission. Specifically, we will aim to (1) improve the theoretical foundation of the work and (2) provide more experimental findings highlighting the utility of the ELSA metric.

Thank you once again for the comments.

The authors.

**Withdrawal Confirmation:**

I have read and agree with the venue's withdrawal policy on behalf of myself and my co-authors.